# AI-enabled alkaline-resistant evolution of protein to apply in mass production

**Liqi Kang[1†], Banghao Wu[2†], Bingxin Zhou[3,4], Pan Tan[5], Yun (Kenneth) Kang[6], Yongzhen Yan[6], Yi Zong[6], Shuang Li[6], Zhuo Liu[3,4,5,7]\*, Liang Hong[1,3,4,5,8]\***

[1]School of Physics and Astronomy, Shanghai Jiao Tong University, Shanghai, China; [2]School of Life Sciences and Biotechnology, Shanghai Jiao Tong University, Shanghai, China; [3]Institute of Natural Sciences, Shanghai Jiao Tong University, Shanghai, China; [4]Shanghai National Centre for Applied Mathematics (SJTU Center), MOE-LSC, Shanghai Jiao Tong University, Shanghai, China; [5]Shanghai Artificial Intelligence Laboratory, Shanghai, China; [6]Changchun GeneScience Pharmaceuticals Co., Ltd., Jilin, China; [7]Department of Biochemistry and Molecular Biology, School of Basic Medical Sciences, Harbin Medical University, Harbin, China; [8]Zhangjiang Institute for Advanced Study, Shanghai Jiao Tong University, Shanghai, China

## eLife Assessment

This **important** work demonstrates the application of Pro-PRIME, a large language model, to engineer VHH antibodies with enhanced stability for extreme industrial environments. The evidence is **convincing**, showing through two rounds of design and experimental validation that AI-guided approaches can outperform traditional rational design methods. The **solid** methodology and results establish a foundation for further exploration of LLM-assisted protein engineering.

**\*For correspondence:**
judeliu@sjtu.edu.cn (ZL);
hongl3liang@sjtu.edu.cn (LH)

[†]These authors contributed equally to this work

**Abstract** Artificial intelligence (AI) models have been used to study the compositional regularities of proteins in nature, enabling it to assist in protein design to improve the efficiency of protein engineering and reduce manufacturing cost. However, in industrial settings, proteins are often required to work in extreme environments where they are relatively scarce or even non-existent in nature. Since such proteins are almost absent in the training datasets, it is uncertain whether AI model possesses the capability of evolving the protein to adapt extreme conditions. Antibodies are crucial components of affinity chromatography, and they are hoped to remain active at the extreme environments where most proteins cannot tolerate. In this study, we applied an advanced large language model (LLM), the Pro-PRIME model, to improve the alkali resistance of a representative antibody, a VHH antibody capable of binding to growth hormone. Through two rounds of design, we ensured that the selected mutant has enhanced functionality, including higher thermal stability, extreme pH resistance, and stronger affinity, thereby validating the generalized capability of the LLM in meeting specific demands. To the best of our knowledge, this is the first LLM-designed protein product, which is successfully applied in mass production.

## Introduction

Protein engineering, situated at the nexus of molecular biology, bioinformatics, and biotechnology, focuses on the design of proteins to introduce novel functionalities or enhance existing attributes (*Lovelock et al., 2022*; *Lutz and Iamurri, 2018*; *Tokuriki and Tawfik, 2009*). With the exponential growth of biological data and computational power, protein engineering has experienced a significant

shift toward advanced computational methodologies, particularly deep learning, to expedite the design process and unravel complex protein–function relationships (*Jang et al., 2022*; *Li et al., 2023*; *Narayanan et al., 2021*; *Qiu and Wei, 2023*; *Tan et al., 2023*; *Zhou et al., 2024*). However, a significant challenge in industrial protein engineering is designing proteins with inherent resistance to extreme conditions, such as high temperature and extreme pH environments (acidic or alkaline) (*Jaenicke, 1991*; *Pinney et al., 2021*). Unlike proteins in natural ecosystems, those used in industrial processes often encounter harsh physical and chemical conditions, necessitating exceptional resilience to maintain functionality (*Rao et al., 2021*; *Xia et al., 2021*). Previous efforts to enhance protein resistance have often relied on rational design and mutant library screening. These methods are typically labor-intensive, inefficient, and yield limited improvements (*Gülich et al., 2002*; *Linhult et al., 2004*; *Minakuchi et al., 2013*; *Palmer et al., 2008*). Consequently, the industrial demand for proteins resilient to harsh environments poses a notable absence within the training datasets of artificial intelligence (AI) models. Exploring whether AI can achieve the evolution of protein resistance to extreme environments is crucial for broadening protein applications and improving modification efficiency.

Recent advances in large-scale protein language models (LLMs) have enabled zero-shot predictions of protein mutants based on self-supervised learning from natural protein sequences (*Hie et al., 2021*; *Madani et al., 2023*; *Rao et al., 2021*; *Rives et al., 2021*). Although AI-guided protein design has been applied to predict the mutants with greater thermostability and higher activity (*Blaabjerg et al., 2023*; *Li et al., 2022*; *Wang et al., 2022*), it is unexplored whether these models based on the natural protein information can find the mutants that adapt the unnatural extreme environments, such as the alkaline solution with the pH value higher than 13.

Here, we employed an LLM developed by our group, the Pro-PRIME model (*Jiang et al., 2024*), to predict dozens of mutants of a nano-antibody against growth hormone (a VHH antibody), and examined their fitness, including alkali resistance and thermostability, to evaluate their performance under extreme environments. We utilized the Pro-PRIME model to score saturated single-point mutations of the VHH in a zero-shot setting, and selected the top 45 mutants for experimental testing. Some mutants exhibited improved alkali resistance, while others demonstrated higher thermal stability or affinity. Subsequently, we fine-tuned the Pro-PRIME model to predict dozens of multi-point mutations. As a result, we obtained three multi-point mutants with enhanced alkali resistance, higher thermostability, as well as strong affinity to the targeted protein. Also, the dynamic binding capacity (DBC) of the selected mutant did not show significant decline after more than 100 cycles, making it suitable for practical application in industrial production. The selected mutant has been used in practical production and lower the cost for over 1 million dollars in a year. To the best of our knowledge, this is the first protein product developed by an LLM that has been successfully applied in mass production. Due to the Pro-PRIME model's ability to achieve precise predictions of multi-point mutations with reliance on a small amount of experimental data, our two-round design process involved experimental validation of only 65 mutants in 2 months, demonstrating remarkable high efficiency. Furthermore, we performed a systematic analysis of these findings and determined that the model can yield more valuable predictive outcomes while remaining consistent with rational design principles. Specifically, within the framework of multi-point combinations, the model's incorporation of negative single-point mutations into the combinatorial space led to exceptional results, showcasing its capacity to capture epistatic interactions. Notably, in striving for global optimum, deep learning methods offer distinct advantages over traditional rational design approaches.

## Results
### Wet experimental testing of single-point mutants

Currently, the prevailing approach in protein engineering involves rational design or high-throughput screening to identify positive single-point mutations, followed by their combination into more effective multi-point mutations through greedy search methods (*Weiß et al., 2018*). However, these approaches are inclined to trap in a local optimum and unable to avoid the negative epistasis (*Khersonsky et al., 2018*; *Weinreich et al., 2006*). To overcome these limitations, we employed the Pro-PRIME model to screen single-point mutations in the first round, enabling rapid inference of scores for saturated mutations. An efficient strategy involves validating only a select number of top-ranked mutants, thereby saving considerable time and reducing economic costs compared to rational design

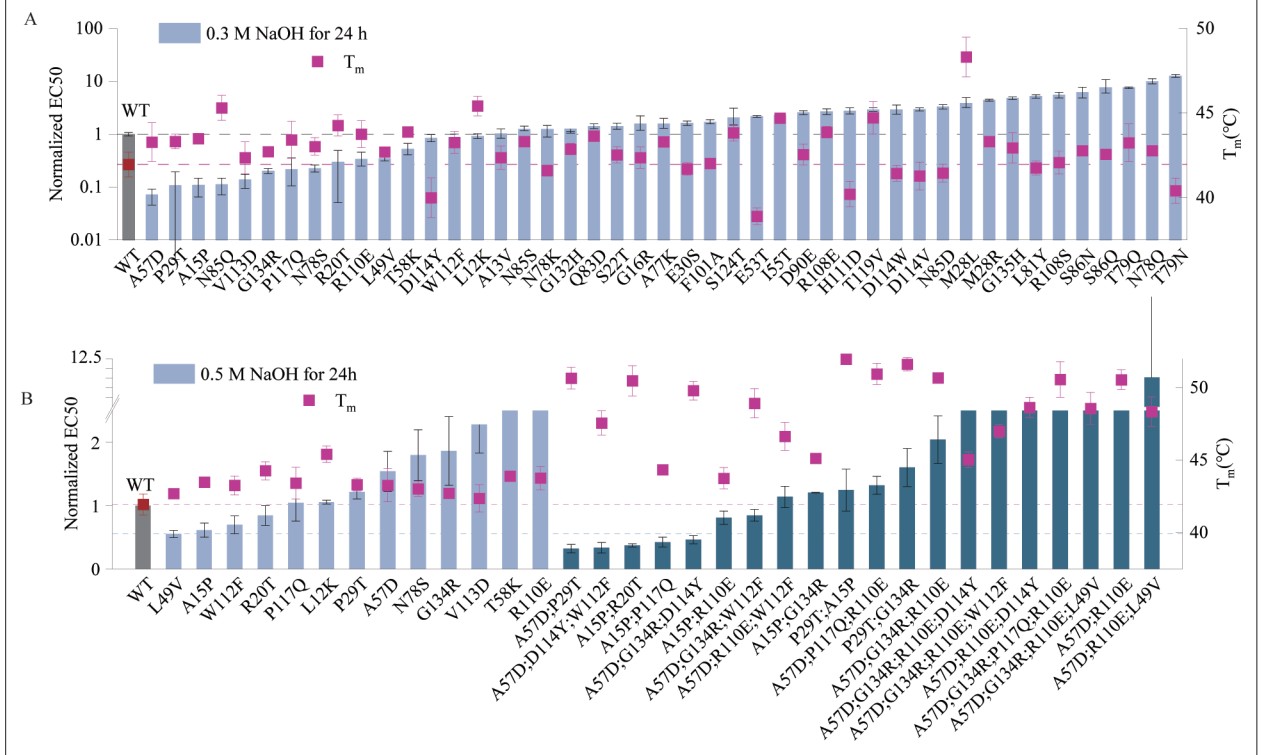

**Figure 1.** The experimental results from two rounds of design. (**A**) Experimental results of single-point mutants. The melting temperatures of mutants are shown as red squares. The affinity of mutants after treatment with 0.3 M NaOH for 24 hr are shown as blue bars. (**B**) Experimental results of multi- and single-point mutants. The melting temperatures of mutants are shown as red squares. The affinity of single-point mutants after treatment with 0.5 M NaOH for 24 hr is shown as blue bars, while those of multi-point mutants are shown as green bars. The affinity values were normalized in the graph, with the wild-type EC50 set as 1. Error bars for EC50 values represent the 95% confidence interval, while error bars for the melting temperatures represent the standard deviation from three independent experiments.

The online version of this article includes the following source data and figure supplement(s) for figure 1:

**Source data 1.** Original data corresponding to *Figure 1A, B*.

**Figure supplement 1.** Experimental results of single-point mutants before alkali treatment.

**Figure supplement 1—source data 1.** Original data corresponding to *Figure 1—figure supplement 1*.

**Figure supplement 2.** Experimental results of multi- and single-point mutants before alkali treatment.

**Figure supplement 2—source data 1.** Original data corresponding to *Figure 1—figure supplement 2*.

**Figure supplement 3.** Distribution plot of the normalized EC50 values after 0.3 M NaOH treatment and $T_m$ values of single-point mutants.

and high-throughput experiment. Moreover, we posit that deep learning models can capture nuanced features during training, and recommend single-point mutations that expert experience may not cover. These mutants expand possibilities for subsequent combination of multi-point mutations and mitigate the risk of converging into local optima.

Specifically, we made use of the Pro-PRIME model to score the single-site saturated mutants, and selected the top 45 single-point mutants of the VHH antibody for testing. Pro-PRIME is a deep learning-based methodology developed to guide protein engineering, utilizing masked language modeling (MLM) and multi-task learning to study and comprehend the semantic and grammatical features inherent in protein sequences, and further capture the temperature traits associated with these sequences. A higher score corresponds to a greater appearance probability of the residue at this site in the Pro-PRIME model (*Jiang et al., 2024*). The alkali resistance of a mutant was determined by its half maximal effective concentration (EC50) after treatment with alkali. A lower EC50 represents that antibodies have stronger affinity. First, the 45 single-point mutants were treated with 0.3 M NaOH for 24 hr, and evaluated by their thermal stability and affinity. The results revealed that 15 mutants exhibited higher alkali resistance, and 35 mutants displayed higher melting temperatures ($T_m$) (*Figure 1A*). We also noted that 8 out of 45 single-site mutants designed by Pro-PRIME

showed improved affinity as compared to the wild type (WT) before alkali treatment (*Figure 1—figure supplement 1*). It is worth noting that six mutants (A57D, A15P, V113D, P117Q, R20T, and L12K) exhibited enhancements in all three properties, that is, higher alkali resistance, greater thermal stability, and stronger affinity before alkali treatment. Additionally, eight mutants (P29T, N85Q, G134R, N78S, R110E, L49V, T58K, and W112F) exhibited improved alkali resistance and $T_m$, albeit at the moderate cost of affinity. This trade-off is acceptable, as excessively high affinity could complicate the separation of growth hormone from the VHH. Furthermore, we subjected single-point mutants with remarkable alkali resistance to treatment with 0.5 M NaOH, and measured their affinity after 24 hr. Our findings demonstrated that four mutants (L49V, A15P, W112F, and R20T) exhibited higher affinity than the WT following the same treatment protocol (*Figure 1B*). Interestingly, these four single-point mutations all showed improvements in $T_m$, and their affinities before alkali treatment were either similar to or lower than those of the WT. However, this does not imply that enhancing thermal stability will concurrently improve alkali resistance. The Spearman correlation between the EC50 values after 0.3 M NaOH treatment and $T_m$ values of single-point mutants is −0.29, indicating that these two properties of the antibody are only weakly correlated (*Figure 1—figure supplement 3*). Therefore, achieving multi-point mutants that enhance both alkali resistance and thermal stability remains a challenge.

## Multi-point mutation design driven by experimental data

In the process of combining multiple single-site mutations, common greedy algorithms typically proceed by sequentially adding the most effective single-point mutations and gradually increasing the number of stacked mutations. However, this approach often demonstrates low efficiency and is susceptible to becoming trapped in local optima. Importantly, the most effective multi-point mutations in practical scenarios may not necessarily include the most effective single-point mutations (*Weinreich et al., 2006*). Moreover, in the traditional approach of incrementally stacking mutations from single to multiple points, only paths that yield better results at each step are retained (*Khersonsky et al., 2018*). Nevertheless, combinations that are discarded due to decreased effectiveness may significantly enhance performance in subsequent stacking step, a phenomenon unforeseeable in traditional methods.

Although the model inference is fast, it is not feasible to explore all possible mutations when designing multi-point mutants due to the exponential increase in the number of potential combinations. To manage this challenge, we constructed a mutant library based on a two-stage design process. In the first stage, we scored all single-point mutations using the model, and in the second stage, we combined experimentally validated single-point mutations to create the multi-point mutant library. This approach ensures that even when designing multi-point mutants (e.g., five-point mutants), the number of mutants to score remains in the millions, which is computationally efficient and practical. The number of single-point mutations selected for the multi-point mutant library is a key factor influencing both the computational load and the scope of the design space. To maintain a balance between efficiency and accuracy, we limited the number of single-point mutations to between 30 and 50. This strategic approach allows us to achieve both scalability and precision in our protein engineering tasks.

To better search the global optimium, we chose the Pro-PRIME model to predict the results of multi-point mutants due to its remarkable performance on the design of diverse proteins (*Jiang et al., 2024*). We trained two Pro-PRIME models using the alkali resistance and $T_m$ data of single-point mutants, respectively, to score these two properties of mutants. During the prediction of multi-point mutations, we prioritized mutants with higher alkali resistance scores while ensuring $T_m$ scores were not lower than those of the WT. Ultimately, we selected 20 multi-point mutants for experimentation (see details in Materials and methods). Experimental results indicated that the alkali resistance of five multi-point mutants (A57D;P29T, A57D;D114Y;W112F, A15P;R20T, A15P;P117Q, and A57D;G134R;D114Y) surpassed that of the best single-point mutant. The $T_m$ of multi-point mutants was generally higher, with the highest (P29T;A15P) exceeding that of the WT by approximately 10°C (*Figure 1B*). Although we did not optimize for the affinity of the VHH antibody, certain multi-point mutants (A15P;R20T, A15P;G134R, and P29T;G134R) exhibited affinity levels close to or even exceeding those of the WT (*Figure 1—figure supplement 2*). Although experimental results from single-point mutations suggested that simultaneously improving the alkali resistance and thermal stability of VHH might

be challenging, we successfully designed multi-point mutants that balance multiple properties. This demonstrates the excellent multi-objective optimization capability of the Pro-PRIME model.

Note that we employed different strategies for designing single- and multi-point mutants, specifically using a zero-shot approach for single-point mutations and fine-tuning the model for multi-point mutations. These choices were made based on the distinct characteristics of the two tasks and the availability of experimental data. For single-point mutations, the number of possible mutations is relatively limited, and at the outset, there were no experimental data available. In such cases, the zero-shot setting was chosen because it allows the model to predict the fitness of mutants based solely on the information learned during pre-training on a large protein sequence dataset. Since single-point mutations are computationally manageable, this approach was deemed appropriate to generate initial predictions for protein engineering. However, when designing multi-point mutants, the situation changes significantly. The potential combinations of mutations increase exponentially, and without prior data, it becomes computationally infeasible to evaluate every possible combination within a reasonable timeframe. Moreover, by the time we reached the multi-point mutation design stage, experimental data for several single-point mutations had already been obtained. This data enabled us to fine-tune the model to better capture the specific structural and functional features that contribute to protein stability and resistance, especially in the context of multiple interacting mutations. Fine-tuning improves the model's accuracy by adjusting its parameters to align more closely with the experimental data, ensuring that the predicted multi-point mutants are more likely to meet the desired engineering goals. After the second round of design, the fitness of the mutants was further improved. In improving alkali resistance, experimental results showed that 15 of the 45 designed mutants exhibited positive responses, yielding a success rate of 30%, close to the 35% success rate achieved in the second round. Compared to the WT, the best single-point mutant improved alkali resistance by approximately 44.7%, while the best multi-point mutant achieved a 67.7% increase. For thermal stability enhancement, the success rate in the first round was 77.8%, rising to 100% in the second round. The top single-point mutant exhibited a $T_m$ increase of 6.37°C over the WT, while the best multi-point mutant had a $T_m$ increase of 10.02°C. We also tested the performance of the zero-shot approach for multi-point mutants, and the results showed that this method did not yield satisfactory predictions. The Spearman correlation coefficient between the zero-shot predictions and experimental results for multi-point mutants was −0.71, indicating a significant discrepancy. This further highlights the importance of fine-tuning the model for multi-point mutations, as the fine-tuned model provided more accurate and reliable results. In summary, the choice of zero-shot for single-point mutations and fine-tuning for multi-point mutations was driven by practical considerations regarding computational feasibility and the availability of experimental data. Fine-tuning the model significantly enhances its predictive performance, particularly for complex multi-point mutations where multiple residues interact. We believe this strategy strikes an optimal balance between computational efficiency and predictive accuracy, making it well suited for practical protein engineering applications.

## Complex epistatic effects generated by the combination of single-point mutations

To understand the principle of the evolution, we analyzed the improvement in multi-point mutants across three different indicators. As illustrated in *Figure 2B*, the evolution of thermal stability in the second round was successful, with nearly all multi-point mutants exhibiting an increase in $T_m$ compared to the best single-point mutant they contain. Three multi-point mutants demonstrated higher affinity than the WT, but only P29T and G134R exhibited positive epistasis, resulting in the creation of a double-point mutant with superior affinity. Although A15P;R20T and A15P;G134R showed higher affinity than the WT, these combinations did not contribute to affinity improvement because the A15P single-point mutation they contained exhibited stronger affinity compared to these double-point mutants.

As shown in *Figure 2A*, single-point mutations produced complex results during the combination process due to epistatic effects, which were unpredictable using traditional methods. A57D;P29T exhibited remarkably high alkali resistance, despite A57D and P29T being negative mutants individually. This 'double negative yields positive' phenomenon also occurred in terms of affinity, where the EC50 value of P29T;G134R was approximately one-third of that of the WT, despite both P29T and G134R caused a decrease in affinity individually, with the EC50 value of P29T being larger than that of

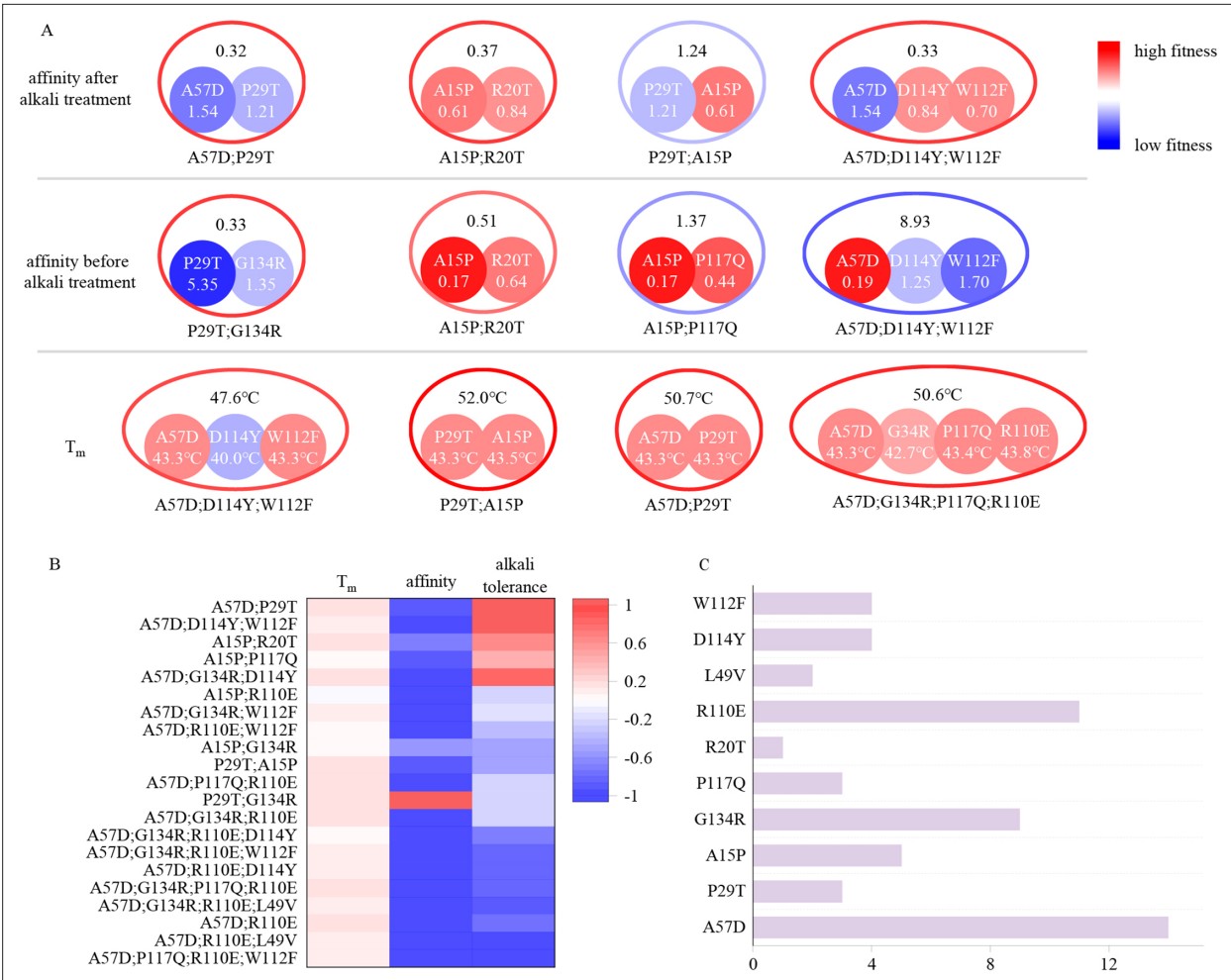

**Figure 2.** Schematic diagram illustrating the combined effects of single-point mutations. (**A**) The colors (outer contours representing multi-point mutations and inner solid circles representing the included single-point mutations) and numerical values represent the EC50 values after alkali treatment with 0.5 M NaOH for 24 hr, the EC50 values before alkali treatment, and the $T_m$ values. Blue indicates mutations inferior to the wild type, while red indicates mutations superior to the wild type. The EC50 values before and after alkali treatment are normalized, with the wild type set to 1. (**B**) Experimental evaluation of multi-point mutations. Different colors represent the proportions of improvement or decline in $T_m$, affinity, and alkali resistance of multi-point mutations compared to the corresponding best single-point mutations they include. (**C**) Distribution of the occurrences of single-point mutations included in the 20 multi-point mutation variants. The length of the bar represents the frequency of occurrence of each mutation.

the WT by a factor of five. These mutation pairs are distantly located on the VHH antibody structure, making it challenging to infer the reasons for the enhanced properties after combination (*Figure 3B*). Not all combinations resulted in improved mutant effects. For instance, while A15P showed high affinity and alkali resistance, adding R20T or P29T led to decrease in affinity or alkali resistance, respectively. Additionally, combining A15P with P117Q, two single mutations capable of individually enhancing affinity, resulted in a decrease in affinity.

In traditional methods, only positive single-point mutations are selected to compose the multi-point mutants and negative single-point mutations are typically avoided in multi-point combinations. However, the number of negative mutations often far exceeds that of positive mutations, thus limiting the sequence space that can be explored by traditional methods. LLMs can leverage negative mutations to generate positive multi-point mutants, surpassing the capabilities of rational design and significantly expanding the design space in protein engineering. Therefore, the Pro-PRIME method manifests significant advantages in exploring sequence space, being less susceptible to local optima, and having greater potential to find the global optimum. As depicted in *Figure 2C*, the 20 multi-point mutations we identified consisted of combinations of nine different single-point mutations. Although the distribution was biased, we incorporated some negative mutations that experts might consider

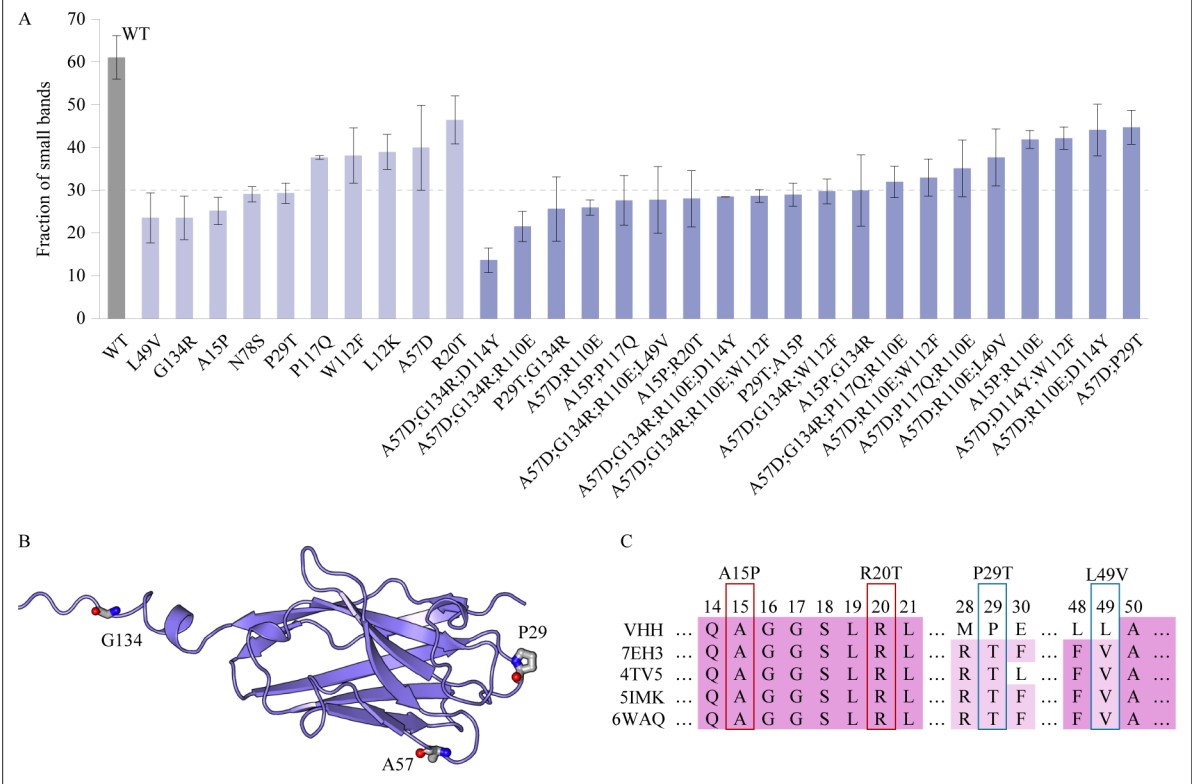

**Figure 3.** Characterization of the VHH antibody, including SDS-PAGE analysis, structural representation, and sequence alignment. (**A**) The SDS–PAGE experimental results depict the proportion of small bands observed after alkali treatment for multi-point mutations and certain single-point mutations exhibiting relatively higher alkali resistance. Error bars for EC50 values represent the 95% confidence interval. (**B**) Structure of the VHH antibody. (**C**) The multiple sequence alignment of the VHH and several homologous sequences.

The online version of this article includes the following source data and figure supplement(s) for figure 3:

**Source data 1.** Original data corresponding to *Figure 3A*.

**Figure supplement 1.** Stability analysis of the wild type (WT) and the A57D;P29T mutant based on molecular dynamics (MD) simulations.

**Figure supplement 1—source data 1.** Original data corresponding to *Figure 3—figure supplement 1B, C, E, F*.

disadvantageous, implying that the Pro-PRIME method balances the exploitation of local optima and the exploration of global optimum.

## Reevaluating the AI predictions from a rational design perspective

In addition to interpret the multi-point mutations based on the single-site mutations, we investigated the design of VHH antibodies from a rational design perspective to understand the AI predictions. The experimental results from SDS–PAGE revealed a significant degradation of the VHH antibody following alkali treatment, with small bands accounting for up to 61.1%. This phenomenon could be attributed to the poor alkali resistance of the VHH antibody. Mass spectrometry analysis identified specific breakage sites at Q2, Q4, G9, G10, G16, S22, S23, S26, A57, N78, N85, F101, and G134, highlighting these as ideal targets for rational design. Introducing mutations at these sites to prevent the breakdown of VHH antibodies could effectively enhance their alkali resistance. The predictions of the Pro-PRIME model align with this rationale, with 4 out of the top 10 performing single-point mutations occurring at these breakpoint sites. Experimental evidence showed that our single-point mutations could effectively reduce the proportion of small bands, and not all positive mutations locate at the breakage sites, such as P29T and A15P (*Figure 3A*). Mutations identified by the model at other locations may have non-local effects, contributing to overall protein stability. In the second round of experiments, multi-point mutants exhibited an overall lower proportion of small bands, reaching as low as 13.6% (A57D;G134R;D114Y), though we did not use the experimental results of SDS–PAGE to train the model and make predictions. Thus, the improvement in the second round suggests a positive

correlation between the alkali resistance of the VHH antibody and the degree of breakage after alkali treatment. Employing a rational design approach, it is plausible to identify single-point mutants with improved alkali resistance at these breakpoints. However, determining the optimal mutated residues requires conducting multiple single-point saturation mutation experiments, which is time consuming and costly. Deep learning methods can directly predict mutation types at each site, enhancing design efficiency while remaining aligned with rational design principles.

Furthermore, we analyzed the VHH antibody and homologous sequences from the perspective of evolutionary sequence alignment (*Figure 3C*). The results revealed that some critical single-point mutations (P29T and L49V) changed original residues to more conservative ones, aligning with the 'consensus sequence' method commonly used in rational design (*Sternke et al., 2019*). The Pro-PRIME model's ability to predict such mutations is expected, given its training set contains information from homologous proteins. However, the advantage of deep learning methods extends beyond providing suggestions aligned with rational design principles; they can predict mutations contrary to expert experience. For instance, both A15 and R20 are conservative residues according to homologous sequence alignment (see *Figure 3C*). Moreover, mutating A15 to proline reduces hydrogen bonds, and changing R20 to threonine contradicts empirical physicochemical properties, since arginine is more likely to form salt bridges and hydrogen bonds, benefiting stability. However, both A15P and R20T contribute to increased alkali resistance and play positive roles in subsequent multi-point combinations. This exceptional capability beyond the traditional rational design principles expands the exploration space of deep learning models, aiding to approach the global optimum.

In order to gain deeper insights into the mechanisms by which the identified mutations enhance protein properties, we performed molecular dynamics (MD) simulations on the best alkali-resistant mutant. The simulation results revealed several key observations that help explain the observed improvements in protein stability and alkali resistance. As shown in *Figure 3—figure supplement 1A*, the two-point mutant of A57D;P29T has a $T_m$ increase of around 8°C and a much stronger binding affinity than the WT. Our analysis of the MD trajectories indicates that the A57D;P29T mutant has a more rigid structure than that of WT due to its lower root-mean-squared deviation (RMSD) of protein (*Figure 3—figure supplement 1B*). Furthermore, we calculated the root-mean-squared fluctuation (RMSF) for each residue, and realized that the mutant displayed less fluctuation at residue 29 but similar flexibility at residue 57. Interestingly, residues at positions 10, 108, and 118 which spatially distant from residues 29 and 57 in the mutant exhibited remarkable weakened fluctuations than those in the WT (*Figure 3—figure supplement 1C*), implying a more rigid structure of the mutant contributing to its improved resistance on high temperature and strong alkalinity. However, *Figure 3—figure supplement 1D* shows the AlphaFold3 predicted structures of the WT and the mutant are quite similar. To unveil the origin of change on structural flexibility, we computed the intramolecular interactions, such as salt bridges and hydrogen bonds for both WT and the mutant. We observed that the mutations increased the number of hydrogen bonds between the mutation sites and the rest of the protein (*Figure 3—figure supplement 1E*). However, the overall structure of the mutant did not show significant changes, which is also evident from the solvent-accessible surface area analysis (*Figure 3—figure supplement 1F*). We also analyzed changes in salt bridges and found that although residue 57 mutated to histidine, no new salt bridges were formed. Additionally, RMSF results showed that residues 10, 108, and 118 became more rigid, but further analysis revealed that there were no significant changes in hydrogen bonds or other interactions in these regions. Taken together, these findings suggest that the enhanced alkali resistance of the mutant is likely due to an overall increase in protein stability, rather than a dramatic change in its structural conformation. The MD simulation results, which are detailed in *Figure 3—figure supplement 1*, provide a deeper understanding of how specific mutations can improve protein properties and offer valuable insights for future protein engineering applications.

## Improved acid and salt resistance simultaneously

According to the rational design principles, a protein with enhanced intramolecular interactions can resist various stress, including high temperature, strong alkali, strong acid, concentrated salt, etc. (*Wijma et al., 2013*; *Xu et al., 2020*). Hence, we expected that the mutants with higher thermal stability and alkali resistance can tolerate acidic and saline environments as well. We evaluated the binding affinity of the mutants to the target protein under elution with saline (1 M NaCl) or acidic

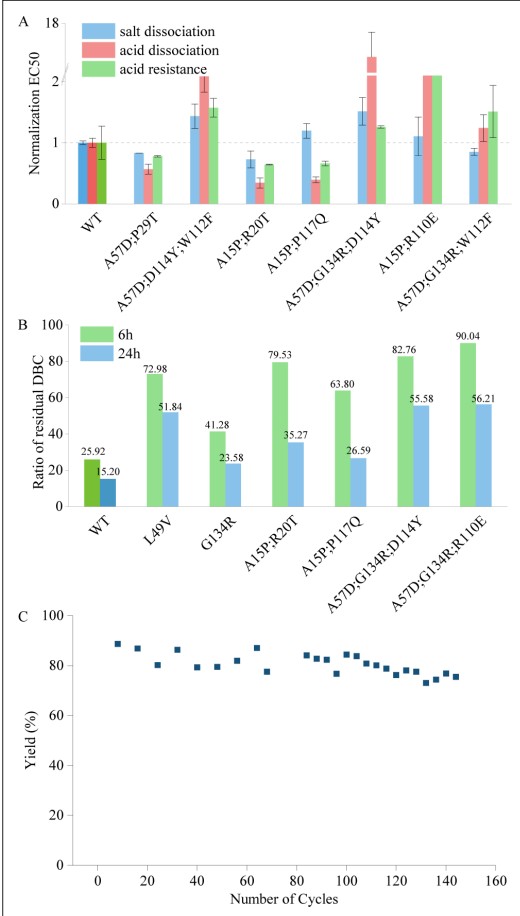

**Figure 4.** Performance of mutants in assays assessing features outside the model's evolutionary predictions. (**A**) The tolerance of acid dissociation and salt dissociation, and the acid resistance of multi-point mutations with strong alkali resistance. All EC50 values are normalized, with the wild type set to 1. Error bars for EC50 values represent the 95% confidence interval. (**B**) Experiment results of residual dynamic binding capacity (DBC) at 10% breakthrough. The bars represent the ratio of the residual DBC after 0.5 M NaOH treatment for 6 and 24 hr to the residual DBC before treatment for multi-site mutants and the wild type. (**C**) Yield in affinity chromatography and the corresponding number of cycles. This figure illustrates the variation in yield during affinity chromatography across multiple cycles.

The online version of this article includes the following source data for figure 4:

**Source data 1.** Original data corresponding to *Figure 4A–C*.

(20 mM citric acid) solutions to characterize their salt- and acid-induced dissociation abilities. Additionally, we assessed the acid resistance of the mutants by measuring their affinity after a 48-hr treatment with 1 M ethanoic acid (see details in Materials and methods). *Figure 4* shows the acid and salt resistance of the selected multi-point mutants with higher thermal stability and stronger alkali resistance. It is noteworthy that the Pro-PRIME model can provide us the mutants, such as A57D;P29T and A15P;R20T, with strong resistance on acid and salt (*Figure 4A*) as well as high temperature and alkali resistance (*Figure 2B*). Our design enhances protein stability while retaining other properties such as affinity, thereby endowing the mutant with the potential for application in industrial production.

## Application in industrial production

The Pro-PRIME method can predict multi-site mutants using a small amount of experimental data, and the entire design process can be completed within 2 months. This method has the potential to empower industrial production due to its relatively low economic and time costs. Our mutants have already been widely applied in the purification process of growth hormones. *Figure 4B* shows the ratio of the residual DBC after 0.5 M NaOH treatment for 6 and 24 hr to the residual DBC before treatment for various mutants and the WT. The WT experienced a 74.1% loss in the residual DBC after 6 hr of alkali treatment, and only retained 15.2% of its residual DBC after 24 hr of alkali treatment. In contrast, the designed multi-site mutants retained 60–90% of their residual DBC after 6 hr of alkali treatment, and even after 24 hr of treatment, some mutants maintained over 50% of their residual DBC, which is remarkably more stable than the WT in the alkaline environment of industrial production. *Figure 4C* shows the variation in yield during affinity chromatography across multiple cycles after employing our mutant. After more than 140 cycles, the yield of our mutant did not exhibit a significant downward trend. In contrary, the binding affinity of WT is unable to sustain after 60 cycles. Therefore, our designed mutants maintain activity after more cycles of reuse compared to the WT, substaintially reducing the production

cost of growth hormones. Consequently, the selected mutant designed by the LLM has been applied in mass production scale up to 5000 l.

## Discussion

Through two rounds of evolution, we successfully designed a VHH antibody with strong resistance to extreme environments and enhanced affinity using the Pro-PRIME model. Although rare case can tolerate the extreme pH and saline conditions in our pre-training dataset, the Pro-PRIME model showed impressive performance after supervised learning with limited data, especially on capturing the epistatic effects. The analysis of these 65 mutants revealed that the Pro-PRIME model is adept at exploring the large space of protein fitness, being less susceptible to local optima, and having greater potential to find the global optimum. Our efficient method of designing mutants that consider multiple properties improvement holds promise for industrial application of proteins. Specifically, the VHH antibody has been deployed in practical production and significantly enhancing the efficiency of the entire production line after our design. While the Pro-PRIME model itself has been reported (*Jiang et al., 2024*), this work demonstrates its first-time application to the challenge of designing proteins with alkali resistance and other extreme properties that are not found in natural proteins, nor have previous studies addressed or provided data for such applications. This shift from optimizing existing protein properties to engineering entirely new, unnatural traits is a significant advance in the field. This study shows that the AI models, such as Pro-PRIME, can not only guide the evolution of protein thermal stability, enzymatic activity, and ligand affinity but also enable to develop the mutants adapting the harsh unnatural environments, such as extreme pH and concentrated salt, largely expanding its application. The novelty of this work lies in the ability to design and engineer proteins with novel properties, specifically alkali resistance, which is an unprecedented achievement in AI-assisted protein engineering. The great potential of AI model is expected to significantly accelerate the development of proteins for diverse applications in medicine, agriculture, bioengineering, etc.

## Materials and methods
### Prediction with Pro-PRIME model

Pro-PRIME is a deep learning-based methodology developed to guide protein engineering, utilizing a comprehensive dataset comprising 96 million sequence-host bacterial strain optimal growth temperatures (*Jiang et al., 2024*). Through MLM and multi-task learning, Pro-PRIME can learn and comprehend the semantic and grammatical features inherent in protein sequences, and further capture the temperature traits associated with these sequences. As a result, Pro-PRIME naturally correlates higher scores with sequences that are more likely to contribute to robustness and survivability in varied environmental conditions, including extreme temperature scenarios. First, we deployed homologous sequences of the protein as an unsupervised dataset, optimizing both the encoder and MLM modules of Pro-PRIME, since additional unsupervised fine-tuning of the language modeling module on homologous protein sequences of target proteins yields improved results. Then we utilized the Pro-PRIME model to predict zero-shot scores for saturated single-point mutations of VHH antibodies, selecting the top 45 for the first round of experiments. When predicting multi-point mutations, we initially established a library of multi-point mutants based on the results of single-point experiments. We selected 20 single-point mutations for combination into multi-point mutants: A57D, P29T, A15P, N85Q, V113D, G134R, P117Q, N78S, R20T, R110E, L49V, T58K, D114Y, W112F, L12K, A13V, N85S, N78K, G132H, and Q83D, as they exhibited strong alkali resistance in preliminary experiments. Regarding supervised learning, the experimental data of the WT and all single-point mutations were used as the training set for the Pro-PRIME model. Alkali resistance and thermal stability were used to train two separate Pro-PRIME models, which predicted the alkali resistance scores and thermal stability scores of mutants, respectively. When screening multi-point mutants, we prioritized mutants with higher alkali resistance scores while requiring their thermal stability scores not to be lower than that of the WT. Although we measured the affinity of proteins in experiments, we did not use this indicator for training the model or screening mutants because excessively high affinity is disadvantageous for purifying growth hormone in practical application.

### MD simulations

The initial structures for MD simulations of both the WT and the mutant were predicted using Alpha-Fold3 (*Abramson et al., 2024*). To simulate experimental conditions, each protein was placed in a cubic water box containing 0.1 M NaCl. The CHARMM27 force field (*Bjelkmar et al., 2010*) and the

TIP4P water model were applied throughout the simulations. After an initial energy minimization of 50,000 steps, the systems were heated and equilibrated for 1 ns in the NVT ensemble at 300 K followed by an additional 1 ns in the NPT ensemble at 1 atm. The production phase then involved 200-ns simulations with periodic boundary conditions, using a 2-fs integration time step. The LINCS algorithm was used to constrain covalent bonds involving hydrogen atoms, while Lennard–Jones interactions were cut off at 10 Å (*Hess et al., 2008*). Electrostatic interactions were computed with the particle mesh Ewald method, using a 10-Å cutoff and a grid spacing of approximately 1.6 Å with a fourth-order spline (*Essmann et al., 1995*). Temperature and pressure were regulated by the velocity rescaling thermostat and Parrinello–Rahman algorithm, respectively (*Bussi et al., 2007*; *Parrinello and Rahman, 1981*). All simulations were performed using GROMACS 2020.4 software packages (*Abraham et al., 2015*). Both systems have reached equilibrium according to the analyses of RMSD.

## Plasmid construction and protein expression

A codon-optimized version of VHH antibody and variants genes were synthesized by Sangon Biotech (Shanghai, China). It was cloned into the pET29(a) plasmid to construct pET29a-VHH-MX with an N terminal His-tag. The expression plasmid was transformed into *E. coli* BL21(DE23) cells. A single colony of each recombinant *E. coli* strain was inoculated into 30 ml LB medium with 50 µg/ml kanamycin for seed culture at 37°C for 12–16 hr. The seed culture was transferred 10 ml to 1 l LB medium with 50 µg/ml kanamycin at 37°C 220 rpm until the OD600 value reached 0.6–0.8. The culture was cooled to 16°C and then induced with 0.5 mM IPTG (isopropyl β-D-thiogalactoside) for 20–24 hr at 16°C.

## Protein purification

Cells were harvested from the fermentation culture by centrifugation for 30 min at 4000 rpm, and the cell pellets were collected for later purification. The cell pellets were resuspended in buffer A (20 mM $Na_2HPO_4$ and $NaH_2PO_4$, 0.5 M NaCl, pH 8.0) and lysed using ultrasonication. The lysates were centrifuged at 12,000 rpm at 4°C for 30 min, after which the supernatants were subjected to Ni-NTA affinity purification with elution buffer (20 mM $Na_2HPO_4$ and $NaH_2PO_4$, 0.5 M NaCl, 250 mM imidazole, pH 8.0). The purity of the fractions obtained was analyzed using SDS–PAGE. The fractions containing the purified target protein were combined and desalted using an ultrafiltration unit. The concentrated protein was finally stored in buffer A supplemented with 10% glycerol at −80°C for long-term preservation.

## Affinity test (ELISA)

Ninety-six-well plates were coated with growth hormone protein at a density of 5 ng/well at 4°C overnight. The plates were washed with 1× PBS'T three times. Following blocking with 1% BSA in 1× PBS at 25°C for 2 hr. After washing three times with 1× PBS'T, the plates were incubated with serial dilutions of VHH proteins 100 µl/well (1:2, 1:4, 1:8, 1:16, 1:32, 1:64, 1:128, 1:256, 1:512, 1:1024, 1:2048) for 1 hr at 25°C. After washing three times with 1× PBS'T, 100 µl/well HRP-labeled Goat Anti-Mouse IgG(H+L) (1:5000) were added and incubated at 25°C for 1 hr. The plates were washed with 1× PBS'T four times, a total of 100 µl/well 3,3′,5,5′-tetramethylbenzidine was added and incubated at 25°C for 15 min in the dark. Finally, 100 µl/well 2 M $H_2SO_4$ was added to stop the reaction and absorbance was measured at 450 nm (TECAN, Swiss).

The log(agonist) versus response – variable slope (four parameters) curves were analyzed to calculate EC50 which determines the stability of VHH after alkaline treatment.

## Differential scanning fluorimetry

The thermal stability testing was carried out using a PCR instrument (Analytik Jena qTower3). All proteins were diluted in 1× PBS to a final concentration of 0.3 mg/ml and mixed with SYPRO Orange at a final concentration of 5× in an eight-row PCR tube. The protein unfolding process was initiated by subjecting the samples to a thermal treatment ranging from 25 to 85°C (with a temperature increment of 0.5°C/step) with each step holding for 5 s. Subsequently, the thermal unfolding curves were obtained, and the data were analyzed using the Boltzmann equation to determine the melting temperature ($T_m$).

## Alkaline pH stability test (SDS–PAGE)

The VHH antibodies (0.5 mg/ml) were treated an equal volume of 0.5 M NaOH for 24 hr, followed by pH adjustment using the same volume of 0.5 M HCl. After centrifugation, the supernatant was

collected and analyzed by SDS–PAGE, and the gray intensity of the fragmented bands was analyzed using ImageJ to determine the alkaline cleavage status of the VHH antibodies.

## Alkaline pH stability test (ELISA)

The VHH antibodies (1.5 mg/ml) were treated an equal volume of 0.3 or 0.5 M NaOH for 24 hr, followed by pH adjustment using the same volume of 0.5 M HCl, the supernatant was collected after centrifugation.

Ninety-six-well plates were coated with growth hormone protein at a density of 5 ng/well at 4°C overnight. The plates were washed with 1× PBS'T three times. Following blocking with 1% BSA in 1× PBS at 25°C for 2 hr. After washing three times with 1× PBS'T, the plates were incubated with serial dilutions of VHH proteins 100 µl/well (1:2, 1:4, 1:8, 1:16, 1:32, 1:64, 1:128, 1:256, 1:512, 1:1024, 1:2048) for 1 hr at 25°C. After washing three times with 1× PBS'T, 100 µl/well HRP-labeled Goat Anti-Mouse IgG(H+L) (1:5000) were added and incubated at 25°C for 1 hr. The plates were washed with 1× PBS'T four times, a total of 100 µl/well 3,3',5,5'-tetramethylbenzidine was added and incubated at 25°C for 15 min in the dark. Finally, 100 µl/well 2 M $H_2SO_4$ was added to stop the reaction and absorbance was measured at 450 nm (TECAN, Swiss).

The log(agonist) versus response – variable slope (four parameters) curves were analyzed to calculate EC50 which determines the stability of VHH after alkaline treatment.

## Acid dissociation test (ELISA)

Ninety-six-well plates were coated with growth hormone protein at a density of 5 ng/well at 4°C overnight. The plates were washed with 1× PBS'T three times. Following blocking with 1% BSA in 1× PBS at 25°C for 2 hr. After washing three times with 1× PBS'T, the plates were incubated with serial dilutions of VHH proteins 100 µl/well (1:2, 1:4, 1:8, 1:16, 1:32, 1:64, 1:128, 1:256, 1:512, 1:1024, 1:2048) for 1 hr at 25°C. After washing three times with 20 mM citric acid and one time with 1× PBS'T, 100 µl/well HRP-labeled Goat Anti-Mouse IgG(H+L) (1:5000) were added and incubated at 25°C for 1 hr. The plates were washed with 1× PBS'T four times, a total of 100 µl/well 3,3',5,5'-tetramethylbenzidine was added and incubated at 25°C for 15 min in the dark. Finally, 100 µl/well 2 M $H_2SO_4$ was added to stop the reaction and absorbance was measured at 450 nm (TECAN, Swiss).

The log(agonist) versus response – variable slope (four parameters) curves were analyzed to calculate EC50 which determines the stability of VHH after acid dissociation.

## Salt dissociation test (ELISA)

Ninety-six-well plates were coated with growth hormone protein at a density of 5 ng/well at 4°C overnight. The plates were washed with 1× PBS'T three times. Following blocking with 1% BSA in 1× PBS at 25°C for 2 hr. After washing three times 1× PBS'T, the plates were incubated with serial dilutions of VHH proteins 100 µl/well (1:2, 1:4, 1:8, 1:16, 1:32, 1:64, 1:128, 1:256, 1:512, 1:1024, 1:2048) for 1 hr at 25°C. After washing three times with 20 mM PB buffer (20 mM $Na_2HPO_4$ and $NaH_2PO_4$, 1 M NaCl, pH 8.0) and one time with 1× PBS'T, 100 µl/well HRP-labeled Goat Anti-Mouse IgG(H+L) (1:5000) were added and incubated at 25°C for 1 hr. The plates were washed with 1× PBS'T four times, a total of 100 µl/well 3,3',5,5'-tetramethylbenzidine was added and incubated at 25°C for 15 min in the dark. Finally, 100 µl/well 2 M $H_2SO_4$ was added to stop the reaction and absorbance was measured at 450 nm (TECAN, Swiss).

The log(agonist) versus response – variable slope (four parameters) curves were analyzed to calculate EC50 which determines the stability of VHH after salt dissociation.

## Acid resistance test (ELISA)

The VHH antibodies (1.5 mg/ml) were treated an equal volume of 1 M ethanoic acid for 48 hr, followed by pH adjustment using the same volume of 1 M NaOH, the supernatant was collected after centrifugation.

Ninety-six-well plates were coated with growth hormone protein at a density of 5 ng/well at 4°C overnight. The plates were washed with 1× PBS'T three times. Following blocking with 1% BSA in 1× PBS at 25°C for 2 hr. After washing three times with 1× PBS'T, the plates were incubated with serial dilutions of VHH proteins 100 µl/well (1:2, 1:4, 1:8, 1:16, 1:32, 1:64, 1:128, 1:256, 1:512, 1:1024, 1:2048) for 1 hr at 25°C. After washing three times with 1× PBS'T, 100 µl/well HRP-labeled Goat

Anti-Mouse IgG(H+L) (1:5000) were added and incubated at 25°C for 1 hr. The plates were washed with 1× PBS'T four times, a total of 100 μl/well 3,3',5,5'-tetramethylbenzidine was added and incubated at 25°C for 15 min in the dark. Finally, 100 μl/well 2 M $H_2SO_4$ was added to stop the reaction and absorbance was measured at 450 nm (TECAN, Swiss).

The log(agonist) versus response – variable slope (four parameters) curves were analyzed to calculate EC50, which determines the stability of VHH after acid treatment.

## Alkaline incubation of VHH followed by liquid chromatography–mass spectrometry analysis

The VHH antibodies (1.5 mg/ml) were treated an equal volume of 0.5 M NaOH for 24 hr, followed by pH adjustment using the same volume of 0.5 M HCl, the supernatant was collected for liquid chromatography–mass spectrometry (LC–MS) after centrifugation. The LC–MS analyses were performed using a Waters Xevo G2 Q-TOF MS instrument (Waters, Milford, MA, USA) equipped with an AQUITY H-class UPLC system (Waters). The MS instrument was calibrated daily with sodium iodide standard provided by the vendor (2 μg/μl in 50:50 2-propanol:water, 700001593). All data were collected using a Zorbax 300SB-C8 (2.1 × 50 mm 1.7 μm) column (Agilent Technologies, Santa Clara, CA, USA). The column was operated at 60°C and 5 μl of each sample was injected (approximately 100–150 nmol on column). The LC gradient consisted of buffer B (0.1% formic acid in water) and buffer C (0.1% formic acid in 80:20 acetonitrile:2-propanol). The gradient was 0–2 min 100% B, 2–10 min 100–10% B, 10–11 min 0% B, 11–12 min 100% B at a flow rate of 0–11 min 0.5 ml/min and 11–12 min 1 ml/min. The MS was run in positive polarity and sensitivity mode at a $m/z$ range of 50–2000, scan rate of 1 spectrum/second and ESI potential set at 3 kV. Determination of molecular weight for each peak in the chromatograms was performed by the provided MaxEnt1 deconvolution function in the MassLynx software (Waters).

## Acknowledgements

This work was supported by the National Natural Science Foundation of China (12204302), the Computational Biology Program of Shanghai Science and Technology Commission (23JS1400600), the Shanghai Pujiang Program (Grant No. 22PJ1406900), the Startup Fund for Young Faculty at SJTU (SFYF at SJTU), the Oceanic Interdisciplinary Program of Shanghai Jiao Tong University (Project No. SL2022MS018), the Natural Science Foundation of Shanghai (Grant No. 23ZR1431700), Shanghai Jiao Tong University Scientific and Technological Innovation Funds (21X010200843), Science and Technology Innovation Key R&D Program of Chongqing (CSTB2022TIAD-STX0017), the Student Innovation Center at Shanghai Jiao Tong University, and Shanghai Artificial Intelligence Laboratory.

## Additional information

### Competing interests

Yun (Kenneth) Kang, Yongzhen Yan, Yi Zong, Shuang Li, Zhuo Liu: Employee of Changchun GeneScience Pharmaceuticals Co., Ltd. The other authors declare that no competing interests exist.

### Funding

| Funder | Grant reference number | Author |
| --- | --- | --- |
| National Natural Science Foundation of China | 12204302 | Zhuo Liu |
| Computational Biology Program of Shanghai Science and Technology Commission | 23JS1400600 | Liang Hong |
| Shanghai Pujiang Program | 22PJ1406900 | Zhuo Liu |
| Startup Fund for Young Faculty at SJTU | SFYF at SJTU | Zhuo Liu |

| Funder | Grant reference number | Author |
|---|---|---|
| Oceanic Interdisciplinary Program of Shanghai Jiao Tong University | SL2022MS018 | Zhuo Liu |
| Natural Science Foundation of Shanghai | 23ZR1431700 | Zhuo Liu |
| Shanghai Jiao Tong University Scientific and Technological Innovation Funds | 21X010200843 | Liang Hong |
| Science and Technology Innovation Key R&D Program of Chongqing | CSTB2022TIAD-STX0017 | Liang Hong |

The funders had no role in study design, data collection and interpretation, or the decision to submit the work for publication.

## Author contributions

Liqi Kang, Conceptualization, Methodology, Writing – original draft, Data curation; Banghao Wu, Methodology, Data curation; Bingxin Zhou, Pan Tan, Writing – review and editing; Yun (Kenneth) Kang, Yongzhen Yan, Yi Zong, Shuang Li, Data curation; Zhuo Liu, Liang Hong, Supervision, Writing – review and editing

## Author ORCIDs

Liqi Kang (iD) https://orcid.org/0009-0004-2427-5807
Bingxin Zhou (iD) https://orcid.org/0000-0002-3897-9766
Zhuo Liu (iD) https://orcid.org/0000-0001-9202-0516
Liang Hong (iD) https://orcid.org/0000-0003-0107-336X

Joint Public Review: https://doi.org/10.7554/eLife.102788.3.sa1
Author response https://doi.org/10.7554/eLife.102788.3.sa2

# Additional files

## Supplementary files

MDAR checklist

## Data availability

All data generated or analyzed during this study are included in the manuscript and supporting files; source data files have been provided for Figures 1, 3, and 4.

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
