## [Editor Report · eLife Assessment]

This **important** work demonstrates the application of Pro-PRIME, a large language model, to engineer VHH antibodies with enhanced stability for extreme industrial environments. The evidence is **convincing**, showing through two rounds of design and experimental validation that AI-guided approaches can outperform traditional rational design methods. The **solid** methodology and results establish a foundation for further exploration of LLM-assisted protein engineering.

---

## [Referee Report · Joint Public Review]

Summary:

In this manuscript, the model's capacity to capture epistatic interactions through multi-point mutations and its success in finding the global optimum within the protein fitness landscape highlights the strength of deep learning methods over traditional approaches.

Strengths:

It is impressive that the authors used AI combined with limited experimental validation to achieve such significant enhancements in protein performance. Besides, the successful application of the designed antibody in industrial settings demonstrates the practical and economic relevance of the study. Overall, this work has broad implications for future AI-guided protein engineering efforts.

Reviewing Editor's comments on revised version:

The authors extensively addressed conceptual and methodological points raised by reviewers, as well as constructive comments to clarify the narrative. Consequently, the manuscript experienced a qualitative jump on clarity and appeal for the eLife readership.

---

## [Author Response]

The following is the authors’ response to the original reviews.

**Reviewer #1 (Public review):**
(1) Summary:In this manuscript, the model's capacity to capture epistatic interactions through multi-point mutations and its success in finding the global optimum within the protein fitness landscape highlights the strength of deep learning methods over traditional approaches.

We thank the reviewer for his/her recognition of our model’s potential and advantages.

(2) Strengths:It is impressive that the authors used AI combined with limited experimental validation to achieve such significant enhancements in protein performance. Besides, the successful application of the designed antibody in industrial settings demonstrates the practical and economic relevance of the study. Overall, this work has broad implications for future AI-guided protein engineering efforts.

We are thankful for the editor’s appreciation on our work, especially acknowledged the practical application of our model.

(3) Weaknesses:However, the authors should conduct a more thorough computational analysis to complement their manuscript. While the identification of improved multi-point mutants is commendable, the manuscript lacks a detailed investigation into the mechanisms by which these mutations enhance protein properties. The authors briefly mention that some physicochemical characteristics of the mutants are unusual, but they do not delve into why these mutations result in improved performance. Could computational techniques, such as molecular dynamics simulations, be employed to explore the effects of these mutations?

We thank the reviewer for this good question, which allows us to provide a deeper investigation into the mechanisms by which the mutations significantly enhance the alkali-resistance of proteins. By following the reviewer’s suggestion, we have expanded our analysis by incorporating molecular dynamics (MD) simulations to understand the impact of the mutations. As an example, we focused on the representative alkali-resistant mutant, A57D;P29T, and examined its MD simulation results. As shown in Figure S4A, the two-point mutant of A57D;P29T has a Tm increase of around 8 ℃ and a much stronger binding affinity than the WT. Our analysis of the MD trajectories indicates that the A57D;P29T mutant has a more rigid structure than that of WT due to its lower root mean squared deviation (RMSD) of protein (Figure S4B). Furthermore, we calculated the root mean squared fluctuation (RMSF) for each residue, and realized that the mutant displayed less fluctuation at residue 29 but similar flexibility at residue 57. Interestingly, residues at positions 10, 108 and 118 which spatially distant from residues 29 and 57 in the mutant exhibited remarkable weakened fluctuations than those in the WT (Figure S4C), implying a more rigid structure of the mutant contributing to its improved resistance on high temperature and strong alkalinity. However, Figure S4D shows the AlphaFold3 predicted structures of the WT and the mutant are quite similar.

To unveil the origin of change on structural flexibility, we computed the intramolecular interactions, such as salt bridges and hydrogen bonds for both WT and the mutant. We observed that the mutations increased the number of hydrogen bonds between the mutation sites and the rest of the protein (Figure S4E). However, the overall structure of the mutant did not show significant changes, which is also evident from the solvent-accessible surface area (SASA) analysis (Figure S4F). We also analyzed changes in salt bridges and found that although residue 57 mutated to Histidine, no new salt bridges were formed. Additionally, RMSF results showed that residues 10, 108, and 118 became more rigid, but further analysis revealed that there was no significant change in hydrogen bonds or other interactions in these regions. Overall, the MD results suggest that more hydrogen bonds introduced by the mutations of A57D;P29T stabilize the protein, leading to the enhanced alkali resistance observed in the mutant. These results are now presented in Figure S4 and discussed in detail in the revised manuscript.

Specifically, we have added the following discussion in the main text:

“In order to gain deeper insights into the mechanisms by which the identified mutations enhance protein properties, we performed molecular dynamics (MD) simulations on the best alkali-resistant mutant. The simulation results revealed several key observations that help explain the observed improvements in protein stability and alkali resistance. As shown in Figure S4A, the two-point mutant of A57D;P29T has a Tm increase of around 8℃ and a much stronger binding affinity than the WT. Our analysis of the MD trajectories indicates that the A57D;P29T mutant has a more rigid structure than that of WT due to its lower root mean squared deviation (RMSD) of protein (Figure S4B). Furthermore, we calculated the root mean squared fluctuation (RMSF) for each residue, and realized that the mutant displayed less fluctuation at residue 29 but similar flexibility at residue 57. Interestingly, residues at positions 10, 108 and 118 which spatially distant from residues 29 and 57 in the mutant exhibited remarkable weakened fluctuations than those in the WT (Figure S1C), implying a more rigid structure of the mutant contributing to its improved resistance on high temperature and strong alkalinity. However, Figure S4D shows the AlphaFold3 predicted structures of the WT and the mutant are quite similar. To unveil the origin of change on structural flexibility, we computed the intramolecular interactions, such as salt bridges and hydrogen bonds for both WT and the mutant. We observed that the mutations increased the number of hydrogen bonds between the mutation sites and the rest of the protein (Figure S4E). However, the overall structure of the mutant did not show significant changes, which is also evident from the solvent-accessible surface area (SASA) analysis (Figure S4F). We also analyzed changes in salt bridges and found that although residue 57 mutated to Histidine, no new salt bridges were formed. Additionally, RMSF results showed that residues 10, 108, and 118 became more rigid, but further analysis revealed that there were no significant changes in hydrogen bonds or other interactions in these regions. Taken together, these findings suggest that the enhanced alkali resistance of the mutant is likely due to an overall increase in protein stability, rather than a dramatic change in its structural conformation. The MD simulation results, which are detailed in Figure S4, provide a deeper understanding of how specific mutations can improve protein properties and offer valuable insights for future protein engineering applications.”

And we also included the following content in the SI:

“Molecular Dynamics (MD) simulations

The initial structures for molecular dynamics (MD) simulations of both the wild type and the mutant were predicted using AlphaFold3. To simulate experimental conditions, each protein was placed in a cubic water box containing 0.1 M NaCl. The CHARMM27 force field and the TIP4P water model were applied throughout the simulations. After an initial energy minimization of 50,000 steps, the systems were heated and equilibrated for 1 ns in the NVT ensemble at 300 K followed by an additional 1 ns in the NPT ensemble at 1 atm. The production phase then involved 200-ns simulations with periodic boundary conditions, using a 2 fs integration time step. The LINCS algorithm was used to constrain covalent bonds involving hydrogen atoms, while Lennard-Jones interactions were cut off at 10 Å. Electrostatic interactions were computed with the particle mesh Ewald method, using a 10 Å cutoff and a grid spacing of approximately 1.6 Å with a fourth-order spline. Temperature and pressure were regulated by the velocity rescaling thermostat and Parrinello-Rahman algorithm, respectively. All simulations were performed using GROMACS 2020.4 software packages. Both systems have reached equilibrium according to the analyses of root mean squared deviation (RMSD).”

(4) Additionally, the authors claim that their method is efficient. However, the selected VHH is relatively short (<150 AA), resulting in lower computational costs. It remains unclear whether the computational cost of this approach would still be acceptable when designing larger proteins (>1000 AA). Besides, the design process involves a large number of prediction tasks, including the properties of both single-site saturation and multi-point mutants. The computational load is closely tied to the protein length and the number of mutation sites. Could the authors analyze the model's capability boundaries in this regard and discuss how scalable their approach is when dealing with larger proteins or more complex mutation tasks?

In our prior work, we have demonstrated that our method is applicable to larger proteins as well [Jiang et al., Sci. Adv. 10, eadr2641 (2024)]. For instance, when engineering a protein with 1000 amino acids, inferring the fitness of one million mutants using the model on a single 4090 GPU takes approximately 20 hours. However, it remains infeasible to explore all possible mutations when designing multi-point mutants due to the vast space. To address this challenge, we propose the design of a reliable mutant library. In the first round of experiments, we used the model to score all single-point mutations, and then constructed the multi-point mutant library by combining experimentally tested single-point mutations. In this way, even when designing five-point mutants, we only need to score on the order of millions of mutants, making the inference process time-efficient and fully acceptable. As a result, the number of single-point mutations selected for combination into the multi-point mutant library becomes a crucial parameter that affects both inference time and scope. We limited the number of single-point mutations to between 30 and 50 to strike a balance between efficiency and accuracy.

These results are discussed in the revised manuscript. Specifically, we have added the following discussion at the section 2.2 in the main text:

“Although the model inference is fast, it is not feasible to explore all possible mutations when designing multi-point mutants due to the exponential increase in the number of potential combinations. To manage this challenge, we constructed a mutant library based on a two-stage design process. In the first stage, we scored all single-point mutations using the model, and in the second stage, we combined experimentally validated single-point mutations to create the multi-point mutant library. This approach ensures that even when designing multi-point mutants (e.g., five-point mutants), the number of mutants to score remains in the millions, which is computationally efficient and practical. The number of single-point mutations selected for the multi-point mutant library is a key factor influencing both the computational load and the scope of the design space. To maintain a balance between efficiency and accuracy, we limited the number of single-point mutations to between 30 and 50. This strategic approach allows us to achieve both scalability and precision in our protein engineering tasks.”

**Reviewer #2 (Public review):**
In this paper, the authors aim to explore whether an AI model trained on natural protein data can aid in designing proteins that are resistant to extreme environments. While this is an interesting attempt, the study's computational contributions are weak, and the design of the computational experiments appears arbitrary.

The reviewer’s comments give us an opportunity to further state the novelty of this study. Despite the AI model has been reported in our previous work [Sci. Adv. 10, eadr2641 (2024)], the unnatural physicochemical properties of proteins, to the best of our knowledge, have never been predicted using AI models. Our preceding work [Sci. Adv. 10, eadr2641 (2024)] has demonstrated that the large language model can predict the performances of the mutants on thermostability, catalytic activity, and binding affinity, etc. However, whether the AI models are able to evaluate the unnatural properties of the mutants remains unexplored. Our work has shown that AI models trained on the natural proteins can be used to design the mutants that resistant extreme conditions, such as strong alkalinity, substantially expanding the application of AI for bioengineering. Moreover, our design of the computational experiments was driven by the nature of the task and the availability of experimental data. We employed different strategies for designing single-point and multi-point mutants, specifically using a zero-shot approach for single-point mutations to overcome the challenge of rare data and fine-tuning the model for multi-point mutations to leverage the experimental data of single-point mutations.

(1) The writing throughout the paper is poor. This leaves the reader confused.

The manuscript has been revised accordingly, and we would like to address the reader’s questions if anything is confused.

(2) The main technical issue the authors address is whether AI can identify protein mutations that adapt to extreme environments based solely on natural protein data. However, the introduction could be more concise and focused on the key points to better clarify the significance of this question.

We thank the reviewer for this comment. We have revised the manuscript, particularly the introduction, where we focused on the research questions, methods, and main findings, while removing excessive background information to improve the manuscript’s conciseness and clarity.

“Protein engineering, situated at the nexus of molecular biology, bioinformatics, and biotechnology, focuses on the design of proteins to introduce novel functionalities or enhance existing attributes[1-3]. With the exponential growth of biological data and computational power, protein engineering has experienced a significant shift towards advanced computational methodologies, particularly deep learning, to expedite the design process and unravel complex protein-function relationships[4-9]. However, a significant challenge in industrial protein engineering is designing proteins with inherent resistance to extreme conditions, such as high temperature and extreme pH environments (acidic or alkaline)[17, 18]. Unlike proteins in natural ecosystems, those used in industrial processes often encounter harsh physical and chemical conditions, necessitating exceptional resilience to maintain functionality[19, 20]. Previous efforts to enhance protein resistance have often relied on rational design and mutant library screening. These methods are typically labor-intensive, inefficient, and yield limited improvements[23-26]. Consequently, the industrial demand for proteins resilient to harsh environments poses a notable absence within the training datasets of Artificial Intelligence (AI) models. Exploring whether AI can achieve the evolution of protein resistance to extreme environments is crucial for broadening protein applications and improving modification efficiency.

Recent advances in large-scale protein language models (LLMs) have enabled zero-shot predictions of protein mutants based on self-supervised learning from natural protein sequences. Although AI-guided protein design has been applied to predict the mutants with greater thermostability and higher activity[34-36], it is unexplored whether these models based on the natural protein information can find the mutants that adapt the unnatural extreme environments, such as the alkaline solution with the pH value higher than 13.

Here, we employed a LLM (large language model) developed by our group, the Pro-PRIME model[27], to predict dozens of mutants of a nano-antibody against growth hormone (a VHH antibody), and examined their fitness, including alkali resistance and thermostability, to evaluate their performance under extreme environments.

We utilized the Pro-PRIME model to score saturated single-point mutations of the VHH in a zero-shot setting, and selected the top 45 mutants for experimental testing. Some mutants exhibited improved alkali resistance, while others demonstrated higher thermal stability or affinity. Subsequently, we fine-tuned the Pro-PRIME model to predict dozens of multi-point mutations. As a result, we obtained three multi-point mutants with enhanced alkali resistance, higher thermostability, as well as strong affinity to the targeted protein. Also, the dynamic binding capacity of the selected mutant did not show significant decline after more than 100 cycles, making it suitable for practical application in industrial production. The selected mutant has been used in practical production and lower the cost for over one million dollars in a year. To the best of our knowledge, this is the first protein product developed by a LLM that has been successfully applied in mass production. Due to the Pro-PRIME model's ability to achieve precise predictions of multi-point mutations with reliance on a small amount of experimental data, our two-round design process involved experimental validation of only 65 mutants in two months, demonstrating remarkable high efficiency. Furthermore, we performed a systematic analysis of these findings and determined that the model can yield more valuable predictive outcomes while remaining consistent with rational design principles. Specifically, within the framework of multi-point combinations, the model's incorporation of negative single-point mutations into the combinatorial space led to exceptional results, showcasing its capacity to capture epistatic interactions. Notably, in striving for global optimum, deep learning methods offer distinct advantages over traditional rational design approaches.”

(3) The authors did not develop a new model but instead used their previously developed Pro-PRIME model. This significantly weakens the novelty and contribution of this work.

While it is true that the Pro-PRIME model was previously developed, the novelty and contribution of this work lie in its novel application to design proteins with properties that are not naturally found or are rare in nature. In our original work, the Pro-PRIME model was used to optimize proteins for existing, well-established properties, such as thermal stability, enzymatic activity, and affinity. However, in this study, we extended the model’s capabilities to design proteins that exhibit resilience to extreme environments, such as high pH—properties that are not inherently present in most natural proteins. To our knowledge, no existing model has addressed the challenge of engineering alkali-resistant proteins, nor is there relevant dataset available for training such models.

This shift from optimizing existing characteristics to engineering entirely new properties represents a significant step forward in the field of protein design. By focusing on the design of proteins that can survive and function in harsh, unnatural environments, we have demonstrated the broader applicability of the Pro-PRIME model beyond its initial scope. This expansion of the model's application is a novel contribution that has the potential to accelerate the development of proteins for industrial, agricultural, and biotechnological applications.

Thus, while the Pro-PRIME model itself is not new, its application to the new challenge of engineering proteins with alkali resistance and other novel properties significantly enhances the impact and novelty of this work. Moreover, this work is groundbreaking not only in terms of the model’s novel application but also because no previous studies have specifically targeted alkali resistance or provided data for training models on such extreme properties. Therefore, our approach is unique, marking a new direction in protein engineering.

We have made the following revisions to the conclusions section of the manuscript:

“Through two rounds of evolution, we successfully designed a VHH antibody with strong resistance to extreme environments and enhanced affinity using the Pro-PRIME model. Although rare case can tolerate the extreme pH and saline conditions in our pre-training dataset, the Pro-PRIME model showed impressive performance after supervised learning with limited data, especially on capturing the epistatic effects. The analysis of these 65 mutants revealed that the Pro-PRIME model is adept at exploring the large space of protein fitness, being less susceptible to local optima, and having greater potential to find the global optimum. Our efficient method of designing mutants that consider multiple properties improvement holds promise for industrial application of proteins. Specifically, the VHH antibody has been deployed in practical production and significantly enhancing the efficiency of the entire production line after our design. While the Pro-PRIME model itself has been reported, this work demonstrates its first-time application to the challenge of designing proteins with alkali resistance and other extreme properties that are not found in natural proteins, nor have previous studies addressed or provided data for such applications. This shift from optimizing existing protein properties to engineering entirely new, unnatural traits is a significant advance in the field. This study shows that the AI models, such as Pro-PRIME, can not only guide the evolution of protein thermal stability, enzymatic activity, ligand affinity, etc., but also enable to develop the mutants adapting the harsh unnatural environments, such as extreme pH and concentrated salt, largely expanding its application. The novelty of this work lies in the ability to design and engineer proteins with novel properties, specifically alkali resistance, which is an unprecedented achievement in AI-assisted protein engineering. The great potential of AI model is expected to significantly accelerate the development of proteins for diverse applications in medicine, agriculture, bioengineering, etc.”

(4) The computational experiments are not well-justified. For instance, the authors used a zero-shot setting for single-point mutation experiments but opted for fine-tuning in multiple-point mutation experiments. There is no clear explanation for this discrepancy. How does the model perform in zero-shot settings for multiple-point mutations? How would fine-tuning affect single-point mutation results? The choice of these strategies seems arbitrary and lacks sufficient discussion.

We appreciate the reviewer’s comment regarding the use of zero-shot and fine-tuning settings for single-point and multi-point mutation experiments, and we are grateful for the opportunity to further clarify this aspect of our work.

In the first round of design, we used the zero-shot approach for single-point mutations because the number of possible single-point mutations is limited, and no prior experimental data was available. In the absence of relevant data, the zero-shot approach allows the model to make predictions based on the learned sequence patterns from the pre-trained protein language model. Given that single-point mutations are relatively fewer in number and computationally feasible to evaluate, the zero-shot approach was deemed appropriate for this task.

However, when it comes to designing multi-point mutants, the number of potential combinations increases exponentially, making it computationally impractical to explore all possible mutations in a reasonable timeframe. Furthermore, since we had already obtained some experimental data for single-point mutations in the first round, we fine-tuned the model with this data in the second round to improve the accuracy of predictions for multi-point mutants. Fine-tuning helps the model better capture the specific features that contribute to protein functionality, which are critical when dealing with multi-point mutations where multiple residues interact. This allows the model to produce more reliable and targeted predictions for multi-point mutants, ultimately leading to better design outcomes.

Regarding the model's performance in zero-shot settings for multi-point mutations, we tested this approach, and the results did not align well with the experimental data for multi-point mutants. Specifically, the Spearman correlation coefficient between the zero-shot predictions and experimental results was -0.71, indicating that zero-shot predictions for multi-point mutations were not as accurate as those from the fine-tuned model.

In summary, the choice of using zero-shot for single-point mutations and fine-tuning for multi-point mutations was driven by the nature of the task and the availability of experimental data. Fine-tuning the model improves its predictive performance, especially for more complex multi-point mutation tasks. We have now clarified these choices in the manuscript and have added further discussion on the trade-offs between zero-shot and fine-tuning approaches.

Specifically, we have added the following discussion at the section 2.2 in the main text:

“Note that we employed different strategies for designing single-point and multi-point mutants, specifically using a zero-shot approach for single-point mutations and fine-tuning the model for multi-point mutations. These choices were made based on the distinct characteristics of the two tasks and the availability of experimental data. For single-point mutations, the number of possible mutations is relatively limited, and at the outset, there were no experimental data available. In such cases, the zero-shot setting was chosen because it allows the model to predict the fitness of mutants based solely on the information learned during pre-training on a large protein sequence dataset. Since single-point mutations are computationally manageable, this approach was deemed appropriate to generate initial predictions for protein engineering. However, when designing multi-point mutants, the situation changes significantly. The potential combinations of mutations increase exponentially, and without prior data, it becomes computationally infeasible to evaluate every possible combination within a reasonable timeframe. Moreover, by the time we reached the multi-point mutation design stage, experimental data for several single-point mutations had already been obtained. This data enabled us to fine-tune the model to better capture the specific structural and functional features that contribute to protein stability and resistance, especially in the context of multiple interacting mutations. Fine-tuning improves the model’s accuracy by adjusting its parameters to align more closely with the experimental data, ensuring that the predicted multi-point mutants are more likely to meet the desired engineering goals. After the second round of design, the fitness of the mutants was further improved. In improving alkali resistance, experimental results showed that 15 of the 45 designed mutants exhibited positive responses, yielding a success rate of 30%, close to the 35% success rate achieved in the second round. Compared to the wild type, the best single-point mutant improved alkali resistance by approximately 44.7%, while the best multi-point mutant achieved a 67.7% increase. For thermal stability enhancement, the success rate in the first round was 77.8%, rising to 100% in the second round. The top single-point mutant exhibited a Tm increase of 6.37°C over the wild type, while the best multi-point mutant had a Tm increase of 10.02°C. We also tested the performance of the zero-shot approach for multi-point mutants, and the results showed that this method did not yield satisfactory predictions. The Spearman correlation coefficient between the zero-shot predictions and experimental results for multi-point mutants was -0.71, indicating a significant discrepancy. This further highlights the importance of fine-tuning the model for multi-point mutations, as the fine-tuned model provided more accurate and reliable results. In summary, the choice of zero-shot for single-point mutations and fine-tuning for multi-point mutations was driven by practical considerations regarding computational feasibility and the availability of experimental data. Fine-tuning the model significantly enhances its predictive performance, particularly for complex multi-point mutations where multiple residues interact. We believe this strategy strikes an optimal balance between computational efficiency and predictive accuracy, making it well-suited for practical protein engineering applications.”